# Current Utilization and Research Status of the Herbal Medicine Guibi-Tang and Its Variants for Cognitive Impairment: A Scoping Review

**DOI:** 10.3390/nu17213365

**Published:** 2025-10-26

**Authors:** Gyeongmuk Kim, Han-Gyul Lee, Seungwon Kwon

**Affiliations:** 1Department of Internal Medicine, College of Korean Medicine, Gachon University, Seongnam 13120, Republic of Korea; kkm156@naver.com; 2Department of Cardiology and Neurology, Kyung Hee University College of Korean Medicine, Kyung Hee University Medical Center, Seoul 02447, Republic of Korea; gyulee0614@hanmail.net

**Keywords:** Guibi-tang, Kami-guibi-tang, dementia, cognitive impairment, herbal medicine

## Abstract

**Background/Objectives:** Guibi-tang (GBT) and its variant Kami-guibi-tang (KGBT) are traditional East Asian multi-herb formulas prescribed for memory loss, insomnia, and fatigue. Preclinical data suggest multimodal neuroprotective actions, including cholinergic signaling modulation and activation of the cAMP response element-binding protein (CREB)/extracellular signal-regulated kinase (ERK) pathway; however, clinical evidence for cognitive disorders remains scattered. This scoping review aimed to map the breadth, design characteristics, efficacy signals, and safety profile of GBT and KGBT across the full spectrum of cognitive impairment. **Methods:** Following the Arksey–O’Malley framework and PRISMA-ScR guidelines, seven databases were searched (MEDLINE, Embase, Cochrane Library, China National Knowledge Infrastructure, ScienceON, Scopus, Citation Information by the National Institute of Informatics) from inception to 31 January 2025, for human studies evaluating GBT or KGBT in subjective cognitive decline, mild cognitive impairment (MCI), dementia, or post-stroke cognitive impairment (PSCI). Two reviewers independently screened, extracted, and charted data on study design, participants, interventions, outcomes, and adverse events. **Results:** Fifteen studies met the inclusion criteria—nine randomized controlled trials, one crossover trial, and five observational reports—enrolling 555 participants (age range, 59–87 years). All were conducted in the Republic of Korea, Japan, or China. GBT or KGBT, given as monotherapy or adjunctive therapy for 4 weeks to 9 months, produced modest but consistent improvements in global cognition (Mini-Mental State Examination/Montreal Cognitive Assessment), memory domains, activities of daily living, and neuropsychiatric symptoms across MCI, Alzheimer’s disease, and PSCI cohorts. Reported adverse event rates were comparable to or lower than those of placebo, usual care, or conventional drugs, and no serious treatment-related toxicity was identified. **Conclusions:** Current evidence—although limited by small sample sizes, heterogeneous formulations, short follow-up durations, and regional concentration—indicates that GBT and KGBT are well tolerated and confer clinically meaningful cognitive and functional benefits. Standardized, multicenter, placebo-controlled trials with biomarker end points are warranted to confirm long-term efficacy, clarify mechanisms, and guide integrative clinical use.

## 1. Introduction

Cognitive impairment refers to a decline in mental abilities—such as memory, attention, and reasoning—that is greater than expected for a person’s age. It spans a clinical spectrum from subjective cognitive decline (SCD) and mild cognitive impairment (MCI) to various types of dementia, including Alzheimer’s disease (AD) and vascular dementia (VD) [1,2,3]. In addition, post-stroke cognitive impairment (PSCI) refers to memory and thinking problems that develop after a stroke [4]. These cognitive impairments are increasingly recognized as major health challenges in the context of global population aging [3], with prevalence continuing to rise worldwide. In 2019, an estimated 50 million individuals lived with dementia globally, and this figure is projected to triple to over 150 million by 2050 [5].

Despite this high burden, current therapeutic options remain limited. There are no approved medications that provide clear symptomatic or disease-modifying benefits for MCI [6]. Even for established dementia, conventional pharmacotherapies such as cholinesterase inhibitors and memantine offer only modest symptomatic relief and do not halt disease progression [7]. This limitation has prompted growing interest in adjunctive and alternative therapies, including traditional herbal medicine approaches, to support cognitive function across the spectrum of cognitive impairment—from MCI to dementia.

Guibi-tang (GBT, Gui Pi Tang in Chinese and Kihito in Japanese) is a herbal prescription traditionally used in Traditional East Asian Medicine (TEAM) for conditions such as memory impairment, insomnia, anxiety, and general debility. The base formula consists of 10 medicinal ingredients—Angelicae Radix, Longan Arillus, Ziziphi Semen, Polygalae Radix, Ginseng Radix, Astragali Radix, Atractylodis Rhizoma Alba, Poria Sclerotium cum Pini Radix, Aucklandiae Radix, and Glycyrrhizae Radix et Rhizoma—and is understood in traditional terms to tonify the heart and spleen to improve “heart–spleen deficiency” symptoms such as forgetfulness and fatigue. A modified version, Kami-guibi-tang (KGBT, also known as Jia-Wei-Gui-Pi-Tang in Chinese or Kamikihito in Japanese Kampo medicine), adds ingredients such as Gardenia jasminoides and Bupleurum falcatum to the base formula. These GBT formulations have been used clinically in the Republic of Korea, China, and Japan for managing cognitive weakness, poor memory, insomnia, anxiety, and related symptoms [8].

Preclinical investigations have steadily demonstrated the pro-cognitive and neuroprotective potential of GBT and its variants. In various animal models, these formulations have been shown to improve learning and memory and reduce Alzheimer-like pathology [9], restore novel object recognition performance to wild-type levels [10], and modulate key neurobiological pathways such as cholinergic signaling and hippocampal cAMP response element-binding protein/extracellular signal-regulated kinase (CREB/ERK) activation [11]. Collectively, these findings provide a pharmacological rationale for advancing GBT-based formulations into clinical evaluation for cognitive impairment.

Emerging clinical studies from East Asia suggest potential cognitive benefits. A Korean pilot randomized controlled trial (RCT) in amnestic MCI reported functional and memory improvements [12]. Another pilot RCT in mild AD showed that add-on KGBT was well tolerated and suggested benefits when paired with acetylcholinesterase inhibitors [13]. Two Japanese trials likewise observed modest Mini-Mental State Examination (MMSE) gains in patients with Alzheimer’s disease receiving GBT-based therapy [14,15]. A multicenter, observer-blind study found notable reductions in behavioral and psychological symptoms of dementia with KGBT [16]. A magnetic resonance imaging-based RCT in amnestic MCI detected hippocampal and fusiform blood flow changes after 24 weeks of KGBT, despite no significant cognitive end point differences [17]. Although these findings are encouraging, most trials remain small or lack rigorous placebo controls, underscoring the need for larger, methodologically robust studies.

Despite growing basic and clinical research, no prior review has systematically synthesized the evidence for GBT or KGBT in the context of cognitive impairment as a broad diagnostic category. Because previous studies differ in design, patient populations, and outcome measures, it is difficult to determine the overall strength of the evidence. Therefore, a scoping review was conducted to comprehensively map the current literature on GBT and KGBT for cognitive impairment—including SCD, MCI, dementia, and PSCI. By organizing the available studies, this review aims to clarify what is currently known, identify areas where the evidence is strong, and highlight gaps that remain. This synthesis can help guide future research and support the integration of traditional herbal medicine into the care of individuals with cognitive impairment.

## 2. Materials and Methods

This study employed a scoping review methodology to explore the current landscape of research involving GBT in the context of cognitive impairment. This approach was selected to provide a comprehensive overview of the breadth and characteristics of the existing literature, while identifying key concepts and research gaps that may guide future investigations.

This study was conducted following the Arksey–O’Malley framework [18] (as refined by Levac et al.) and reported in accordance with the PRISMA-ScR (Preferred Reporting Items for Systematic reviews and Meta-Analyses extension for Scoping Reviews) checklist and explanation [19]:(1)Formulation of the research questions;(2)Identification of relevant literature;(3)Selection of eligible studies;(4)Data charting and organization;(5)Synthesis and reporting of findings.

### 2.1. Identifying the Research Questions

Before initiating the review, we formulated the following primary research question:

“What studies have applied GBT or its modified forms to cognitive impairment or dementia?”

The primary aim of this review was to analyze the existing literature on the clinical outcomes of GBT in patients with cognitive impairment. Based on this objective, the following specific research questions were defined:(1)What types of studies have been conducted?(2)What types of cognitive impairment or dementia have been treated with GBT?(3)What improvements have been observed following the use of GBT?(4)What is the level of evidence supporting the efficacy of GBT or its variants in this context?

### 2.2. Literature Search

The search strategy was developed using the PICO framework, focusing on the use of GBT for treating cognitive impairment. The search was restricted to clinical studies involving human participants; preclinical studies conducted on animals or at the cellular level were excluded. The intervention was limited to the traditional herbal formula GBT, as defined within the framework of TEAM. All types of cognitive impairment were considered, regardless of etiology.

Studies utilizing modified forms of GBT (KGBT) were included, as these formulations adjust the original composition based on patient-specific symptoms or conditions. Various dosage forms, including decoctions, pills, powders, and capsules, were accepted without restriction, while non-oral administration methods—such as acupuncture point injection or intravenous infusion—were excluded.

No restrictions were applied to the type of control groups or outcome measures. Eligible studies were limited to clinical research and included case reports, case series, prospective and retrospective observational studies, before-and-after studies, RCTs, integrative reviews, and systematic reviews. Only full-text articles published in peer-reviewed academic journals were considered; unpublished manuscripts and conference abstracts were excluded.

The literature search was conducted across seven electronic databases: MEDLINE (PubMed), Embase, Cochrane Library, China National Knowledge Infrastructure, ScienceON, Scopus, and Citation Information by the National Institute of Informatics. Additionally, relevant studies identified outside the structured search process were manually reviewed and included. A comprehensive search of titles, abstracts, subject headings, and full texts was conducted on 5 February 2025, covering publications up to 31 January 2025, using the search terms detailed in the Appendix A.

For example, a representative PubMed search was (“Guibi-tang” OR “Guibitang” OR “Kihi” OR “Guipi” OR “Gui pi”) AND (“Cognitive” OR “Dementia” OR “Alzheimer”). Full search strategies for all databases are provided in the Appendix A.

### 2.3. Literature Selection

All retrieved citations were imported into EndNote version 21. Two independent reviewers (GK and HL) screened titles and abstracts for relevance. Full texts were obtained for potentially eligible studies and assessed against the predefined inclusion and exclusion criteria. HL performed a secondary check of the initially selected articles. Any discrepancies in study selection were resolved through discussion until consensus was reached; if agreement could not be achieved, a third reviewer (SK) made the final determination. After deduplication, the remaining records were first screened by title and abstract, and then their introductions were examined to confirm eligibility for inclusion in the review.

### 2.4. Data Extraction and Organization

The studies selected for inclusion were examined in relation to the main research questions. Each study’s fundamental characteristics—first author, publication year, academic field, and study design—were recorded. Each paper was then systematically cataloged to include participant details, disease classification, intervention specifics, assessment methods, and primary outcomes. This approach provided a clear overview of the populations studied, the treatments administered, and the reported effects.

Data were derived primarily from observational and interventional studies, with systematic reviews included when relevant. All information was transcribed exactly as presented in the original sources. Authors were identified by the surname of the first author; publication year was recorded as the year indexed in the bibliographic database; and field of study was determined by the discipline of the first author’s institution. Study designs—observational, interventional, or literature review—were classified using the framework outlined in the National Institute for Health and Care Excellence Public Health Guidance (2012) [20]. If a methodology was not explicitly labeled, the most appropriate category was assigned based on the description provided.

A table was prepared to summarize the number and proportion of studies by publication year, design, and academic field. Basic participant information—including total sample size, sex distribution, age range, treatment duration, and disease subtype—was also recorded. When only a sex ratio was reported, participant counts were estimated by applying the ratio to the total sample size. Treatment duration was noted in days or months; if not directly stated, it was calculated from the date of the first intervention to the final treatment date. For studies with both treatment and control groups, details for each group were listed separately. Any missing information was marked as “Not reported,” and disease diagnostic criteria were copied verbatim from the source.

Intervention details included the herbal prescription name, ingredient composition, dosing regimen (doses per day and frequency), and total treatment period. If dosage or frequency was not reported, this was noted as “Not reported.” When the exact formula was not provided, the Korea Pharmaceutical Information Center (www.health.kr) or the original text was consulted. Adjunctive therapies—such as acupuncture, moxibustion, cupping, or standard Western treatments—were also summarized if administered alongside the herbal medicine.

Assessment methods were extracted directly from each study’s methodology. Observational studies were summarized in terms of clinical progression metrics, while interventional studies were documented according to their specified evaluation tools and time points.

Each study’s main findings were recorded as described in its Discussion and Conclusion Sections. When both pre- and post-treatment data were available, both values were reported. For studies that included multiple follow-up assessments, outcomes from the earliest and latest time points were presented to illustrate treatment effects over time.

## 3. Results

### 3.1. Literature Search and Selection Process

By 31 January 2025, the database search yielded 218 publications. Following deduplication, 171 records were removed, and an additional 30 were excluded after title and abstract screening because they did not concern GBT or cognitive impairment, did not involve human subjects, or were unrelated to herbal interventions. Four articles were excluded at the full-text review stage — three due to the unavailability of the original manuscripts and one due to an inappropriate methodology for inclusion. Ultimately, 15 studies [12,13,14,15,16,17,21,22,23,24,25,26,27,28,29] met all inclusion criteria and were carried forward for analysis (Figure 1).

### 3.2. Study Designs and Regions

Of the 15 studies reviewed, 10 were experimental studies [12,13,14,15,16,17,21,22,26,27] and 5 were observational studies [23,24,25,28,29]. Among the experimental studies, nine were RCTs [12,13,14,16,17,21,22,26,27] and one was an open-label crossover clinical trial [15]. The observational studies comprised one retrospective chart review [29] and four case reports [23,24,25,28]. Of the nine RCTs, four targeted patients with dementia [13,14,16,22], four targeted patients with MCI [12,17,26,27], and one enrolled cognitively normal individuals [21]. Regarding disease type, three studies focused on AD [13,14,16], four on PSCI [13,16,17,27], and one on amnestic MCI [26]. One trial included both AD and PSCI in its target population [16]. The included studies were conducted in three countries, with the Republic of Korea accounting for the largest portion (eight of fifteen, 53.3%) [12,13,17,21,23,24,28,29]; Japan contributed four studies (26.7%) [14,15,16,25], and the remaining three randomized controlled trials (20%) were conducted in China [22,26,27] (Table 1).

### 3.3. Demographic Characteristics of Study Participants

After excluding literature review articles, the 15 primary studies together enrolled 555 participants. Of these, 520 (93.7%) were included in RCTs or other controlled trials, and 35 (6.3%) were described in case reports or retrospective chart reviews. Overall, 238 participants (42.9%) were men and 317 (57.1%) were women. Participant ages ranged from 59 to 87 years, with three studies not reporting age [17,22,26] (Table 1).

### 3.4. Details of Intervention

#### 3.4.1. Dosage, Frequency, and Treatment Period of Herbal Medicine

In this context, “dosage” refers to the total daily quantity of GBT ingredients, “frequency” to the number of administrations per 24 h period, and “treatment period” to the total duration of therapy. Of the fifteen included studies, nine (60%) evaluated extract formulations [12,13,14,15,16,17,21,25,29] and six (40%) employed decoction preparations [22,23,24,26,27,28]. When the prescribed amount was considered a single dosage unit, extract studies administered a total of 7.5–9 g per day: Kwak et al. [21] delivered 8.7 g in two divided doses; five studies administered 7.5 g in two divided doses [14,15,16,25,29]; and four studies administered 9 g in three divided doses [12,13,17,29]. In the six decoction studies, one dosage unit per administration was used. Three studies conducted in China [22,26,27] administered one unit twice daily, whereas three case studies used two doses divided into three daily administrations [23,24,28]. Treatment periods ranged from 28 days to 9 months; among interventional studies, durations ranged from 6 to 24 weeks (Table 1).

#### 3.4.2. Composition of GBT, KGBT, and Herbal Medicines Included in Their Variants

The GBT formulations used in the included studies were all modified from the original composition. Beyond these standard modifications, the most frequently incorporated botanicals across clinical studies were Zingiberis Rhizoma and Zizyphi Fructus; ginger appeared in eight trials [12,13,14,15,16,23,24,29] and jujube seed in seven of these same studies [12,13,14,15,16,24,29]. Other commonly added ingredients included Bupleuri Radix and Gardeniae Fructus in four studies [12,13,16,29] and Acori Graminei Rhizoma in four studies [21,23,26,28]. Meanwhile, Saussureae Radix [14,15,16] and Moutan Cortex [12,13,16,27,29] were each featured in three studies, underscoring their supportive yet meaningful roles in tailoring GBT variants for cognitive impairment research (Table 1).

#### 3.4.3. Types of Concomitant Medication

Of the fifteen included studies, six [14,17,21,22,26,27] reported GBT monotherapy without any concomitant medications. The remaining nine studies allowed additional agents as follows: cognitive enhancers (acetylcholinesterase inhibitors and memantine) in four studies [13,15,16,29]; cardiovascular drugs (amlodipine, atenolol, carvedilol, losartan) in three case report studies [24,25,28]; antiplatelet and lipid-lowering therapies (aspirin, cilostazol, pravastatin, ezetimibe, rosuvastatin) in two studies [24,28]; neuromodulators and psychotropics (anxiolytics, hypnotics, anticonvulsants, levetiracetam, quetiapine) in two studies [16,28]; and supplements or other supportive agents (choline alfoscerate, thiamine, gliatilin, gliatamine, nimodipine) in four studies [23,24,26,28] (Table 2).

### 3.5. Evaluation Methods

All studies employed a variety of cognitive and functional assessment instruments, with the MMSE and its regional adaptations emerging as the primary screening tools. The Barthel Activities of Daily Living (ADL) scale or the Korean Modified Barthel Index (K-MBI) were the next most common endpoints, reported in six studies [12,13,14,22,27,28], while structured dementia staging using the Clinical Dementia Rating (CDR) appeared in five studies [12,13,23,24,28]. Hemodynamic changes were assessed in two studies: regional cerebral blood flow (CBF) measured via single-photon emission computed tomography (SPECT) by Higashi et al. [14] and via arterial spin labeling magnetic resonance imaging (ASL-MRI) by Cho et al. [17]. Domain-specific cognitive tests—such as the Montreal Cognitive Assessment (MoCA) [27], Rivermead Behavioural Memory Test (RBMT) [27], Trail Making Test Part B (TMT-B) [27], Seoul Neuropsychological Screening Battery–Dementia (SNSB-D) [12,13], and Memory Assessment Scales (MAS) [21]—were also used in one or two studies. Additional specialized neuropsychological and functional measures included the Geriatric Depression Scale (GDS) [12,13,24,29], Neuropsychiatric Inventory (NPI) [13,16,27], Manual Muscle Testing (MMT) [28], and Videofluoroscopic Swallowing Study (VFSS) [28], each used in one to four studies. When ranked by frequency of use, MMSE variants, ADL scales, and CDR emerged as the core instruments guiding efficacy evaluation in GBT/KGBT cognitive impairment research (Table 2).

### 3.6. Treatment Outcomes by Endpoints

Across the fifteen studies, the most consistently improved endpoint was global cognitive function, with eight trials [12,14,15,17,22,23,28,29] reporting significant increases in MMSE scores or those of its variants. Barthel ADL or K-MBI scores improved in four studies [14,22,27,28]. Dementia severity, staged using the CDR, decreased significantly in four trials [12,23,24,28]. Hemodynamic measures of cerebral perfusion improved in two studies: regional CBF assessed by SPECT [14] and by ASL-MRI [17]. Domain-specific cognitive tests—including MoCA, RBMT, TMT-B, SNSB-D, and MAS—showed significant gains in three studies [12,13,28]. Mood and neuropsychiatric symptoms, assessed by GDS and NPI, improved in three studies [16,27,29]. Functional swallowing, evaluated by VFSS, and motor performance, assessed by MMT, improved in one study (Table 2) [28].

However, not all subdomain outcomes showed consistent benefits. For instance, while Higashi et al. [14] and Yim et al. [29] reported improvements in orientation and attention, other domains such as language or visuospatial function did not change. Shin et al. [12] found that memory scores improved, but other SNSB-D domains and daily function measures remained unchanged. By contrast, Kim et al. [13] observed no significant effects across any domain in mild AD. These findings suggest that GBT/KGBT may exert domain-specific rather than global effects, and that interpretation of biochemical or imaging outcomes should be made cautiously, as their clinical significance remains uncertain.

### 3.7. Treatment Outcomes by Type of Disease

Classified by severity of cognitive impairment, ten studies addressed dementia [13,14,15,16,22,23,24,25,28,29], five focused on MCI [12,17,26,27,29], one investigated SCD [29], and one included participants with normal cognition [21]. When categorized by the nature of cognitive impairment, four studies involved AD [13,14,15,16], six investigated PSCI including VD [16,17,22,24,27,28], one study examined amnestic cognitive impairment [12], and one addressed alcoholic dementia [23]. Nogami et al. [16] included both AD and PSCI, while Yim et al. [29] encompassed a broad spectrum from SCD and MCI to dementia (Table 2). In terms of dementia severity, evidence from 10 studies indicates that GBT and KGBT generally produced favorable clinical effects. Higashi et al. [14] and Watari et al. [15] reported better global cognition—particularly orientation and attention—than their respective control conditions. In post-stroke cohorts, Liu (2012) [16,17,22,24,27,28], along with case reports by Kim et al. [24] and Youn et al. [28], described clearer cognitive and functional recovery relative to comparators or baseline. Behavioral benefits were emphasized by Nogami et al. [16], Lee et al. [23], and Tamano et al. [25], who observed reductions in agitation, anxiety, insomnia, or caregiver burden accompanied by cognitive stability. A nine-month observational study by Yim et al. [29] showed sustained cognitive gains and reduced depressive symptoms, most evident in vascular dementia. Notably, Kim et al. [13] found no cognitive advantage over placebo in patients with mild Alzheimer’s disease, suggesting that efficacy may be more evident in earlier stages such as MCI rather than established AD.

Studies investigating mild cognitive impairment showed the following results: Shin et al. [12] found improved memory-composite and global cognition in amnestic MCI compared with placebo. Cho et al. [17] reported higher MMSE scores and beneficial changes in brain metabolites and perfusion after KGBT in post-stroke patients. Li et al. [27] found GBT variants superior to an active herbal comparator, enhancing cognition, daily living, and neuropsychiatric status while lowering oxidative stress markers. Feng et al. [26] observed symptom relief accompanied by increased acetylcholine levels and decreased homocysteine levels. Collectively, these studies suggest that GBT/KGBT can improve cognition, functional ability, and cerebral physiology in prodromal or post-stroke settings, warranting larger confirmatory trials.

Across four AD studies, GBT and its Kami/Kihito variants produced modest yet clinically relevant effects. Higashi et al. [14] documented global cognitive improvement, particularly in orientation and attention; Watari et al. [15] reported additional MMSE gains when Kihito was added to a stable cholinesterase inhibitor regimen; Kim et al. [13] observed no cognitive advantage in moderate AD but noted better responsiveness at the earlier amnestic mild cognitive impairment (aMCI) stage; and Nogami et al. [16] reported marked reductions in agitation, anxiety, and related behavioral symptoms while cognition remained stable. Across PSCI studies, GBT and KGBT consistently produced clinically notable improvements. Liu [22] reported superior MMSE and ADL improvements compared with piracetam; Cho et al. [17] observed higher K-MMSE scores and favorable brain metabolite changes compared with placebo; Li et al. [27] demonstrated greater MoCA, memory, ADL, and antioxidant improvements than an active comparator; Nogami et al. [16] reported marked reductions in neuropsychiatric symptoms; and case reports by Kim et al. [24] and Youn et al. [28] described recovery in cognition, motor function, ADL, and swallowing.

### 3.8. Safety

Adverse effects were reported in five studies [12,13,14,16,21]. In one RCT [16], two cases of aspiration pneumonia led to treatment discontinuation; no other severe adverse events were reported. Other adverse effects included hypertension, diarrhea, dyspepsia, temporomandibular joint pain, and mild skin rash (Table 2).

## 4. Discussion

This study aimed to examine the current research landscape regarding the use of GBT in managing cognitive impairment. The analysis was guided by four specific questions: (1) What types of research designs have been employed? (2) For which subtypes or conditions related to cognitive impairment has GBT been applied? (3) What therapeutic outcomes have been reported following GBT intervention? (4) What is the strength of the existing evidence regarding the clinical efficacy of herbal medicine interventions?

### 4.1. Research Status

All fifteen studies included in this review were conducted in East Asia—the Republic of Korea (eight studies, 53.3%), Japan (four studies, 26.7%), and China (three studies, 20.0%)—despite the use of global databases in the search process. This regional concentration likely reflects the long-standing clinical application of GBT and its variant KGBT within TEAM settings, where most related clinical research is undertaken.

Of the fifteen studies evaluating GBT for cognitive impairment, nine were RCTs, representing the largest proportion of study designs (60%). These RCTs were conducted across all three countries (Republic of Korea: four; China: three; Japan: two), underscoring a broad interest in rigorous evaluation throughout East Asia. The frequent use of objective cognitive-function measures such as the MMSE and MoCA provides readily quantifiable endpoints, potentially reducing logistical and ethical barriers and making RCT implementation more feasible in cognitive impairment research than in many other clinical conditions.

Across the nine RCTs, three used placebo controls [12,13,17], three employed non-treatment or usual-care arms [14,16,21], two administered GBT as an adjunct to standard pharmacotherapy [22,26], and two [14,27] adopted an alternative herbal medicine comparator. In placebo-controlled trials, both groups exhibited small improvements on broad cognitive screening tools; however, active treatment effects emerged on domain-specific neuropsychological tests and biomarkers. In patients with aMCI, KGBT increased SNSB-D (total and memory) and CDR-SB scores, whereas no between-group difference was observed on the K-MMSE. This suggests that placebo or practice effects may obscure treatment-related changes when using coarse screening measures. Such effects, potentially mediated through dopaminergic and endogenous opioid pathways [30], can transiently enhance motivation and attention, producing modest global score gains within the measurement error range. Nevertheless, clear group separation on measures such as SNSB-D and biomarkers including GABA+/Cr ratios and region-specific CBF supports the presence of pharmacological effects beyond placebo. Among the trials using GBT as an adjunct to conventional pharmacotherapy, one paired it with nimodipine and the other with piracetam. Liu et al. [22] found that, compared with piracetam monotherapy, adjunctive GBT improved cerebral perfusion and metabolism, upregulated nerve growth factor and brain-derived neurotrophic factor (BDNF), and suppressed reactive oxygen species and pro-inflammatory cytokines—likely attributable to its saponins and polysaccharides. These multimodal effects translated into superior cognitive outcomes and higher Barthel Index scores, indicating improved functional capacity. Similarly, Feng et al. [26] proposed that GBT’s polysaccharides and saponins activate the cholinergic system and, unlike nimodipine—which primarily acts via calcium-channel blockade—modulate multiple neuroprotective pathways, thereby alleviating both cognitive deficits and systemic symptoms. Collectively, these findings suggest that GBT enhances conventional pharmacotherapy by exerting central nervous system-directed actions while improving overall health status, leading to better functional performance.

### 4.2. Types of Cognitive Impairment

The included literature showed a marked skew toward advanced stages of cognitive decline. When stratified by severity, two-thirds of studies (10/15) enrolled patients with clinically diagnosed dementia [13,14,15,16,22,23,24,25,28,29], whereas one-third (5/15) targeted MCI [12,17,26,27,29]. Only a single trial each investigated SCD [29] and cognitively normal older adults [21]. This pattern indicates that the evidence base for GBT/KGBT is most developed in symptomatic, late-stage populations, with limited data for prodromal or preclinical phases.

Categorization by etiology revealed further heterogeneity. Four studies focused on AD [13,14,15,16], six examined PSCI, including VD phenotypes [16,17,22,24,27,28], one addressed amnestic MCI [12], and one investigated alcoholic dementia [23]. Notably, Nogami et al. [16] enrolled both AD and PSCI cohorts in parallel, while Yim et al. [29] adopted a spectrum approach encompassing SCD, MCI, and established dementia, reflecting increasing interest in stage-transitional research.

### 4.3. Significance of GBT Treatment

All 15 studies employed standardized cognitive-function measures. Of the 10 interventional studies, most reported statistically significant cognitive improvements in treatment groups compared with controls, most frequently reflected in higher MMSE or MoCA scores. Additionally, four case reports documented recovery toward age-adjusted normative ranges, suggesting clinically meaningful benefits of GBT/KGBT across diverse contexts, including PSCI and alcoholic dementia.

Across the four AD studies, GBT/KGBT demonstrated consistent clinical benefits, although these were often confined to specific cognitive domains. In particular, two studies [13,14] observed modest but statistically significant gains in global cognition—especially in orientation and attention—beyond those achieved with usual care or cholinesterase inhibitors. Nogami et al. [16] reported marked reductions in neuropsychiatric symptoms, notably agitation, dysphoria, anxiety, disinhibition, and irritability, accompanied by improved performance in instrumental ADL. Notably, Kim et al. [13] suggested stage-dependent effects, with cognitive improvements being more pronounced at prodromal stages and diminishing as the disease advances.

Across the four PSCI studies, GBT/KGBT consistently provided multidomain benefits. Two studies [17,22] reported meaningful gains over standard care, including higher MMSE/MoCA scores and faster TMT-B performance, reflecting both overall cognitive recovery and improved executive processing speed. These same studies also documented parallel improvements in ADL, indicating that cognitive gains translated into greater functional autonomy. In another study [16], KGBT markedly reduced Neuropsychiatric Inventory–Nursing Home version scores, alleviating agitation, dysphoria, anxiety, and related symptoms, thereby complementing cognitive benefits. Furthermore, according to findings from a functional MRI study [17], KGBT modestly preserved regional CBF—particularly in memory-related areas such as the hippocampus and fusiform gyrus—in PSCI patients, although metabolite changes were minimal, warranting higher-dose, larger-scale trials.

Regarding safety, reported adverse effects were limited to mild transient gastrointestinal complaints, with no serious adverse events observed. From an experimental research perspective, fermented Guibi-tang exhibited neither acute oral toxicity in rats nor genotoxicity in bacterial mutagenicity, chromosomal aberration, or mouse micronucleus assays. Accordingly, the study concluded that Guibi-tang is safe and non-toxic, supporting its continued application in complementary and alternative medicine [31]. In another experimental study, subchronic oral administration of GBT aqueous extract in rats for 13 weeks produced no mortality or toxicity across clinical, hematological, biochemical, or organ-related parameters. Apart from mild forestomach and duodenal mucosal hyperplasia at the highest dose (5000 mg/kg/day), GBT was considered non-toxic, with a no-observed-adverse-effect level established at 2000 mg/kg/day [32]. Consistent with previous findings, clinical use of GBT was not associated with significant adverse effects. In a study on chronic heart failure [33], the incidence of adverse events did not differ between the GBT and control groups. Similarly, in a study on insomnia [34], the incidence of adverse events was lower in the GBT group compared with the conventional medicine group. Collectively, these findings indicate that GBT has a favorable safety profile, with adverse event rates comparable to or lower than those of control or standard treatments.

Taken together, the available evidence suggests that GBT/KGBT offers clinically meaningful benefits in advanced cognitive impairment, including clinically diagnosed dementia and PSCI. These findings support the rationale for large, multicenter RCTs. By contrast, data for earlier or prodromal stages—such as subjective cognitive decline or cognitively normal but high-risk older adults—remain limited. Therefore, small exploratory studies should first confirm safety and investigate potential disease-modifying or neuroprotective effects before larger trials are undertaken. In addition, future research should include less-studied etiologies, such as Lewy body dementia and frontotemporal dementia, to determine whether the benefits of GBT/KGBT extend to a broader range of pathogenic mechanisms.

### 4.4. Proposed Treatment Mechanisms for GBT in Cognitive Impairment

Preclinical studies provide a pharmacological rationale for the use of GBT and its variants in cognitive impairment. In animal models, GBT demonstrated neuroprotective effects, including improved memory performance, increased synaptic protein expression, and reduced amyloid-β-induced neuronal damage [9]. These findings suggest that the multi-herb formulation may exert pro-cognitive effects by enhancing neurotransmitter function and preserving neural structure. In 5XFAD transgenic mice—a widely used experimental model for Alzheimer’s disease—oral administration of KGBT restored novel-object recognition memory to wild-type levels [10]. KGBT also reversed amyloid-β-induced axonal degeneration by enhancing protein phosphatase 2A activity and reducing tau hyperphosphorylation, both in vitro and in vivo, in 5XFAD models [9]. In another animal experiment, KGBT improved memory and learning in scopolamine-treated mice by up-regulating cholinergic markers and activating CREB/ERK pathways in the hippocampus [11].

Taken together, these findings suggest that GBT/KGBT may improve cognition through several interrelated mechanisms: (1) cholinergic enhancement and synaptic plasticity; (2) modulation of amyloid and tau pathology; (3) antioxidative and anti-inflammatory neuroprotection; and (4) structural and functional support for neural circuits involved in learning and memory. Collectively, these mechanisms may contribute to the cognitive improvements observed in clinical settings, although further translational studies are warranted to confirm their relevance in human populations.

### 4.5. Strength of the Study and Clinical Suggestions

This review distinguishes itself by encompassing a remarkably broad etiologic spectrum. Rather than limiting the analysis to a single disease entity, evidence was collated from studies on aMCI, AD, PSCI, and alcoholic dementia. Synthesizing findings from fifteen studies spanning these diverse conditions provides a comprehensive perspective on the efficacy of GBT/KGBT—one that transcends traditional disease-specific boundaries and highlights its potential relevance across the full continuum of cognitive disorders.

The consistency of cognitive benefits observed across neurodegenerative, vascular–ischemic, and metabolic or toxic pathomechanisms supports the premise that GBT/KGBT exerts core neuroprotective and neuromodulatory effects applicable to “cognitive impairment” as a clinical syndrome rather than to any single diagnosis. Such disease-agnostic utility aligns with the TEAM principle of modifying the general condition of patients, suggesting that the formula’s multifaceted mechanisms—anti-inflammatory, antioxidative, cholinergic modulation, and cerebrovascular enhancement—may converge to support cognition irrespective of the underlying etiology.

Finally, the evidence base demonstrates both monotherapeutic and adjunctive value. Seven of the fifteen included studies administered GBT/KGBT as a standalone treatment and still documented significant improvements in global cognition and/or functional independence. The remaining trials combined GBT/KGBT with conventional agents such as acetylcholinesterase inhibitors, piracetam, or nimodipine, yielding additive or synergistic gains without notable safety concerns. Collectively, these findings indicate that clinicians may flexibly incorporate GBT/KGBT either when standard pharmacotherapies are ineffective or contraindicated or as a complementary intervention to potentiate existing regimens.

### 4.6. Limitations

This scoping review provides a preliminary synthesis of the potential benefits and mechanisms of GBT/KGBT for cognitive impairment; however, several methodological and evidentiary limitations should be acknowledged. First, most included RCTs were single-center and small in scale (mean sample size ≈ 60), which limits statistical power. Additionally, one-third of the included studies comprised observational designs or case reports, in which selection bias and confounding are only partially controlled. Second, there was considerable variation in formulation (decoction vs. extract), daily dosage (7.5–9 g vs. one herbal packet), and treatment duration (4 weeks to 9 months). Another limitation concerns the standardization of GBT and KGBT. Although both formulas share a core set of constituent herbs as described in classical texts, multiple clinical variants exist, reflecting the inherent flexibility of TEAM practice. This heterogeneity, while characteristic of TEAM, poses challenges for reproducibility and standardization. In modern clinical practice, pharmaceutical-grade extracts of GBT/KGBT in the Republic of Korea and Japan are manufactured under standardized conditions with Good Manufacturing Practice (GMP)-certified raw materials, ensuring composition consistency. By contrast, decoction-based preparations inherently show greater variability due to raw material sourcing and the traditional boiling process. Although national-level quality control regulations are enforced in East Asian countries, international harmonization of herbal standards remains limited. These factors should be considered when evaluating the reproducibility and clinical reliability of the current evidence. Many studies also permitted concomitant medications—most notably cholinesterase inhibitors and cardiovascular agents—making it difficult to isolate the intrinsic effects of GBT/KGBT. Third, global cognition was typically assessed with the MMSE, while domain-specific neuropsychological tests and objective biomarkers were rarely employed. Furthermore, no study followed participants for ≥12 months, precluding conclusions about treatment durability or disease-modifying potential. Fourth, all studies originated from East Asia (Republic of Korea, Japan, China), populations with shared genetic and cultural backgrounds. A substantial proportion were published in local, non-English journals, increasing the likelihood of publication bias and limiting generalizability. In addition, cultural practices, dietary habits, and genetic background may have influenced both the tolerability and efficacy of GBT/KGBT, and thus caution is needed when extrapolating these findings to non-East Asian populations.

Well-controlled, multicenter RCTs employing standardized GBT/KGBT formulations and rigorous protocol registration are needed. Future trials should incorporate objective biomarkers—such as cerebrospinal fluid Aβ/tau, serum neurofilament light chain, or arterial spin-labeled MRI cerebral blood flow—and extend follow-up to at least 12 months to determine long-term efficacy and potential neuroprotective effects.

## 5. Conclusions

This scoping review indicates that GBT and its variant KGBT provide modest yet consistent benefits for cognitive impairment—ranging from mild cognitive impairment to Alzheimer’s disease and post-stroke cognitive decline—across fifteen East Asian clinical studies (nine RCTs, one crossover trial, and five observational studies). Reported benefits include improvements in global cognition, memory, activities of daily living, and neuropsychiatric symptoms. Adverse event rates were comparable to or lower than those with placebo, usual care, or standard pharmacotherapy, and no serious toxicity was reported. These findings suggest that GBT/KGBT may be deployed flexibly, either as monotherapy when conventional agents are unsuitable or as a safe adjunct to enhance existing regimens. Nonetheless, the current evidence base is constrained by small sample sizes, formulation heterogeneity, short follow-up durations, and a geographic concentration of studies within East Asia. Large, multicenter, placebo-controlled trials using standardized preparations and incorporating objective biomarker endpoints are required to confirm long-term efficacy, elucidate mechanisms of action, and guide clinical integration.

## Figures and Tables

**Figure 1 nutrients-17-03365-f001:**
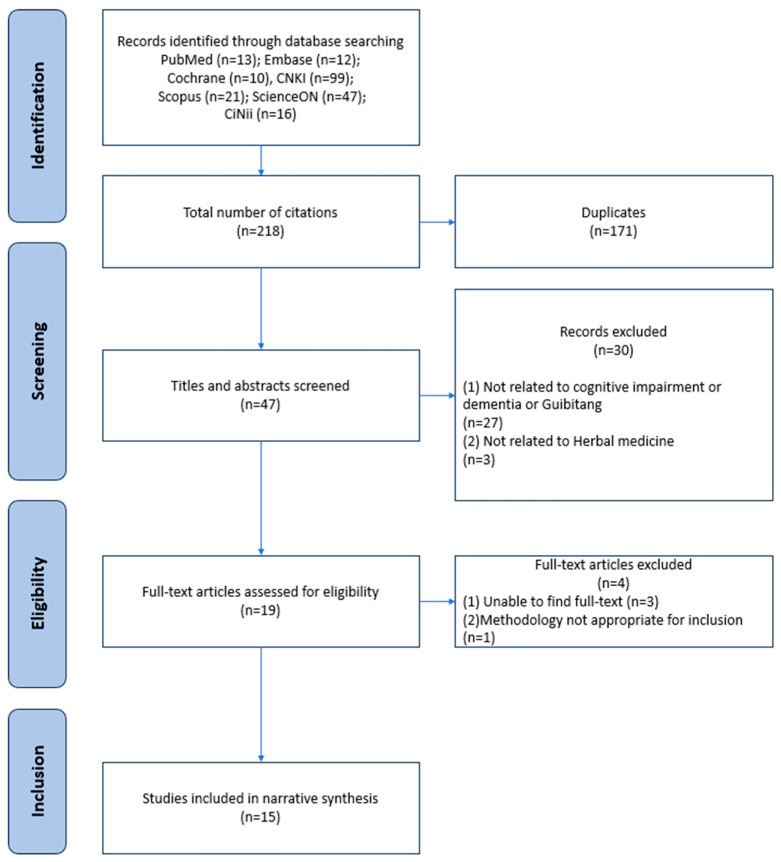
Study flow chart.

**Table 1 nutrients-17-03365-t001:** Summary of included studies on GBT and KGBT for cognitive impairment: demographics, ingredients, dosage, frequency, and treatment duration.

Author	Year	Study Location	Research Design	Sample Size (Male/Female)	Age (Treatment/Control)	Herbal Medicines Added (+) or Removed (–) from the Original Composition of GBT ^†^	Formulation	Dosage and Administration	Treatment Duration
Higashi [14]	2007	Japan	RCT	64 (11/53)	Wuchasingihwan: 84.7 ± 6.5/GBT: 84.3 ± 6.1/Non-treatment: 82.8 ± 8.1	(+): Saussureae Radix, Zingiberis Rhizoma, Zizyphi Fructus (–): Aucklandiae Radix	Extract	7.5 g/day, p.o., TID	3 months
Kwak [21]	2008	Republic of Korea	RCT	63 (13/50)	52.0 ± 7.1/50.8 ± 6.7	(+): Acori Graminei Rhizoma, Schisandrae Fructus, Cnidii Rhizoma, Ginkgo Folium (–): Longan Arillus, Ziziphi Semen, Astragali Radix, Atractylodis Rhizoma Alba, Poria Sclerotium cum Pini Radix, Aucklandiae Radix, Glycyrrhizae Radix et Rhizoma	Extract	8.7 g/day, p.o., BID	6 weeks
Liu [22]	2012	China	RCT	58 (39/19)	Not reported	(+): Codonopsis Radix (–): Atractylodis Rhizoma Alba, Poria Sclerotium cum Pini Radix	Decoction	1 dose/day, p.o., BID	10 weeks
Feng [26]	2021	China	RCT	70 (38/32)	Not reported	(+): Codonopsis Radix, Poria cum Ligno Hospite, Acori Graminei Rhizoma (–): Ginseng Radix, Poria Sclerotium cum Pini Radix	Decoction	1 dose/day, p.o., BID	30 days
Cho [17]	2021	Republic of Korea	RCT	30 (18/12)	Not reported	Not reported	Extract	9 g/day, p.o., TID	24 weeks
Shin [12]	2021	Republic of Korea	RCT	33 (17/16)	70.13 (Treatment: 70.2/Control: 70.1)	(+): Moutan Cortex, Zizyphi Fructus, Zingiberis Rhizoma Recens, Bupleuri Radix, Gardeniae Fructus	Extract	9 g/day, p.o., TID	24 weeks
Li [27]	2022	China	RCT	114 (66/48)	63.91 ± 7.26/63.78 ± 7.47	(+): Persicae Semen, Carthami Flos, Cnidii Rhizoma, Ginkgo Folium, Salviae Miltiorrhizae Radix, Rehmanniae Radix Praeparata, Alpiniae Oxyphyllae Fructus (–): Ziziphi Semen, Poria Sclerotium cum Pini Radix, Aucklandiae Radix	Decoction	1 dose/day, p.o., BID	8 weeks
Kim [13]	2023	Republic of Korea	RCT	14 (3/11)	70.7 ± 6.8/75.7 ± 10.1	(+): Moutan Cortex, Gardeniae Fructus, Zizyphi Fructus, Bupleuri Radix, Zingiberis Rhizoma Recens	Extract	9 g/day, p.o., TID	24 weeks
Nogami [16]	2023	Japan	RCT	63 (18/45)	82.5 ± 6.6/84.0 ± 5.1	(+): Zizyphi Fructus, Zingiberis Rhizoma, Saussureae Radix, Gardeniae Fructus, Bupleuri Radix (–): Aucklandiae Radix	Extract	7.5 g/day, p.o., TID	28 days
Watari [15]	2019	Japan	Open-label crossover clinical trial	10 (4/6)	71.8 ± 6.93	(+): Zizyphi Fructus, Zingiberis Rhizoma, Saussureae Radix (–): Aucklandiae Radix	Extract	7.5 g/day, p.o., TID	24 weeks
Yim [29]	2024	Republic of Korea	Retrospective chart review	31 (9/22)	Overall: 69.6 ± 11.2; AD: 64.5; MCI: 71.2 ± 10.3; SCD: 67.5; VD: 70.5	(+): Moutan Cortex, Zizyphi Fructus, Zingiberis Rhizoma Recens, Bupleuri Radix	Extract	9 g/day, p.o., TID or 7.5 g/day, p.o., TID	≥ 90 days
Lee [23]	2016	Republic of Korea	Case report	1 (1/0)	59	(+): Acori Graminei Rhizoma, Zingiberis Rhizoma Recens	Decoction	2 doses/day, p.o., TID	29 days
Kim [24]	2018	Republic of Korea	Case report	2 (1/1)	83, 66	(+): Zizyphi Fructus, Zingiberis Rhizoma Recens	Decoction	2 doses/day, p.o., TID	(1) 87 days (2) 29 days
Tamano [25]	2018	Japan	Case report	1 (0/1)	82	Not reported	Extract	7.5 g/day, p.o., TID	9 months
Youn [28]	2023	Republic of Korea	Case Report	1 (0/1)	87	(+): Acori Graminei Rhizoma, Zingiberis Rhizoma Recens, Zingiberis Rhizoma Recens	Decoction	2 doses/day, p.o., TID	59 days

^†^ Original composition of GBT: Angelicae Radix, Longan Arillus, Ziziphi Semen, Polygalae Radix, Ginseng Radix, Astragali Radix, Atractylodis Rhizoma Alba, Poria Sclerotium cum Pini Radix, Aucklandiae Radix, Glycyrrhizae Radix et Rhizoma. RCT, randomized controlled trial; p.o., per os (oral administration); BID, bis in die (twice daily); TID, ter in die (three times daily).

**Table 2 nutrients-17-03365-t002:** Summary of the included studies.

Author	Year	Target Disease	Herbal Intervention	Comparator	Concomitant Treatment	Outcome Measures	Key Efficacy Results	Safety/Adverse Events
Higashi [14]	2007	AD	GBT	Wuchasingihwan (control 1), no treatment (control 2)	None	MMSE; Barthel ADL; regional CBF (SPECT)	MMSE score improved: +1.65 ± 0.53 (*p* < 0.01); MMSE orientation sub-score improved vs. non-treatment (*p* = 0.001); MMSE attention sub-score improved vs. non-treatment (*p* = 0.006). Raw MMSE sub-score data not reported.	Hypertension (GBT); diarrhea (control 1)
Kwak [21]	2008	Normal cognition	GBT	No treatment	None	K-MAS; AST, ALT, BUN, creatinine	K-MAS visual delayed recognition improved in control group: +2.4 ± 2.4 (vs. case +0.6 ± 2.7, *p* = 0.023); K-MAS language memory process—word clustering recall improved in case group: +0.1 ± 0.2 (vs. control +0.0 ± 0.1, *p* = 0.017); ALT increased in case group: +7.8 ± 7.5 (vs. control +2.6 ± 5.7, *p* = 0.010).	Adverse event rate: 34% (treatment), 19.5% (control)
Liu [22]	2012	PSCI	GBT	Piracetam	None	MMSE; Barthel ADL	Clinical effectiveness: 75.9% (treatment) vs. 37.9% (control), *p* < 0.05; MMSE improved: 22.47 ± 3.17 vs. 19.39 ± 1.44 (*p* < 0.05); Barthel ADL improved from 50.71 ± 8.5 to 72.47 ± 10.76 (treatment, *p* < 0.05); post-treatment Barthel ADL higher vs. control (*p* < 0.05).	Not reported
Feng [26]	2021	aMCI	GBT variant 1	Nimodipine 30 mg	None	TCM symptom score; ACh; Hcy	TCM score improved (observation < control, *p* < 0.05); ACh increased (*p* < 0.05); Hcy decreased (*p* < 0.05).	Not reported
Cho [17]	2021	PSCI	KGBT	Placebo	None	MRS metabolites (NAA/Cr, Cho/Cr, Glx/Cr, mI/Cr, GABA+/Cr); CBF (ASL-MRI); K-MMSE	K-MMSE difference (F = 4.71, *p* = 0.039); GABA+/Cr difference (F = 5.27, *p* = 0.029); CBF decreased in both groups (cluster 2 *p* < 0.001, hippocampus *p* < 0.001, fusiform gyrus *p* = 0.003); within KGBT, similar decreases; fusiform CBF % decrease lower vs. placebo (*p* = 0.024); cluster 1 CBF higher vs. placebo (F = 8.71, *p* = 0.006).	Not reported
Shin [12]	2021	aMCI	KGBT	Placebo	None (excluded ChEIs, memantine)	s-D total; CDR-SB; SNSB-D memory; SVLT; RCFT; K-MMSE; GDS; Barthel ADL; K-IADL; SGDS	CDR-SB improved in KGBT (*p* = 0.010), worsened in placebo; SNSB-D total and memory improved (*p* < 0.001); no significant changes in K-MMSE, GDS, ADL/IADL, depression.	Adverse event rate: 11.8% (treatment), 12.5% (control)
Li [27]	2022	PSCI	GBT variant 2	Red deer ginseng tablets	None	MoCA; RBMT; TMT-B; Barthel ADL; NPI-1/2; TCM symptom score; serum 8-OHDG, ox-LDL, MDA, SOD, Hcy, IL-8, CRP, FIB	Cognitive improvement rate: 92.98% vs. 78.95% (*p* < 0.05); MoCA recovery ≥ 26: 54.39% vs. 33.33% (*p* < 0.05); post-treatment MoCA, RBMT, ADL higher; TMT-B time shorter (*p* < 0.01); TCM score, NPI-1/2 lower (*p* < 0.01); SOD higher; 8-OHDG, ox-LDL, MDA, Hcy, IL-8, CRP, FIB lower (*p* < 0.01).	Not reported
Kim [13]	2023	AD	KGBT	Placebo + ChEI	ChEI	SNSB-D total; SNSB sub-scores; K-MMSE; CDR; GDS; SGDS; K-IADL; Barthel ADL; KQoL-AD; CGA-NPI	No significant change in SNSB-D total/domains (*p* > 0.6); K-MMSE declined in both (*p* = 0.80); earlier-stage aMCI data showed better effects in memory and executive function; no significant differences in SGDS, ADL/IADL, CDR, GDS.	No adverse events reported
Nogami [16]	2023	AD, PSCI	KGBT	No treatment	ChEIs, memantine, anxiolytics, hypnotics, anticonvulsants	NPI-NH; DEI; MMSE; labs	NPI-NH improved: 29.8 → 13.2 (*p* < 0.001); DEI improved: 24.3 → 32.5 (*p* = 0.001); significant between-group differences in NPI-NH subcategories (agitation, dysphoria, anxiety, disinhibition, irritability).	Aspiration pneumonia (*n* = 2)
Watari [15]	2019	AD	GBT	ChEI	None	MMSE-J; RBANS-J	MMSE-J improved during kihito intake vs. ChEI-only (*p* = 0.0039)	Not reported
Yim [29]	2024	SCD, MCI, AD, VD	KGBT	None	cognitive medications (e.g., donepezil, rivastigmine, memantine); gliatilin, gliatamine, nutritional supplements	MMSE-K (total/subdomains); S-GDS	MMSE-K improved at 3 months (*p* = 0.021) and 9 months (*p* = 0.041); time effect (*p* = 0.007); attention/calculation and orientation improved; VD subgroup improved at 9 months (*p* = 0.015); S-GDS decreased (*p* = 0.006)	Not reported
Lee [23]	2016	Alcoholic dementia	KGBT	None	Donepezil Thiamine	MMSE-K; CDR	MMSE-K: 20 → 25 after 28 days; CDR: 1 → 0.5; orientation, attention/calculation, memory recall improved.	Not reported
Kim [24]	2018	PSCI	KGBT	None	Case 1: Donepezil, Atenolol, Amlodipine, Acetylcysteine, Rebamipide, Aspirin, Choline Alfoscerate, Finasteride, Silodosin Case 2: Aspirin, Cilostazol, Losartan, Pravastatin	MMSE-K; CDR; GDS	Case 1 (83 M): MMSE-K 11 → 21 → 27; sub-items improved; CDR 2 → 2 → 0.5; GDS 6 → 5 → 3. Case 2 (66 F): MMSE-K unchanged, recall improved (0 → 1); CDR 2 → 1; GDS 6 → 5.	Not reported
Tamano [25]	2018	Dementia (unspecified)	KGBT	None	Donepezil Amlodipine Sitagliptin	BPSD; family burden	Week 1: calmer, better sleep; 1 month: reduced wandering, improved orientation/executive function; 2 months: improved appetite, weight gain, anemia recovery, family QOL.	Not reported
Youn [28]	2023	PSCI	GBT variant 4	None	Levetiracetam, choline alfoscerate, carvedilol, amlodipine, metformin, famotidine, sodium alginate, quetiapine, acetaminophen/tramadol, ezetimibe, rosuvastatin, levothyroxine	K-MMSE-2; CDR; MMT; K-MBI; VFSS	K-MMSE-2: 6 → 14; CDR: 3 → 2; MMT grade 2 → 3; K-MBI: 0 → 17; VFSS PAS 8 → 5, then solid food intake; delirium resolved; improved communication/orientation after 2 weeks.	Not reported

AD, Alzheimer’s disease; GBT, Guibitang; MMSE, Mini-Mental State Examination; ADL, activities of daily living; SPECT, single-photon emission computed tomography; K-MAS, Korean Memory Assessment Scales; AST, Aspartate Aminotransferase; ALT, Alanine Aminotransferase; BUN, Blood Urea Nitrogen; PSCI, post-stroke cognitive impairment; aMCI, amnestic mild cognitive impairment; TCM, Traditional Chinese Medicine; ACh, Acetylcholine; Hcy, Homocysteine; KGBT, Kami-guibi-tang; NAA, N-acetylaspartate; Cr, Creatine; Cho, Choline; Glx, Glutamate + Glutamine; mI, Myo-inositol; GABA, Gamma-aminobutyric acid; MRS, Modified Rankin Scale; CBF, cerebral blood flow; ASL-MRI, arterial spin labeling MRI; K-MMSE, Korean Mini-Mental State Examination; SNSB-D, Seoul Neuropsychological Screening Battery for Dementia; CDR-SB, Clinical Dementia Rating–Sum of Boxes; SVLT, Seoul Verbal Learning Test; RCFT, Rey–Osterrieth Complex Figure Test; GDS, Geriatric Depression Scale; K-IADL, Korean Instrumental Activities of Daily Living; SGDS, Short Geriatric Depression Scale; MoCA, Montreal Cognitive Assessment; RBMT, Rivermead Behavioural Memory Test; TMT-B, Trail Making Test Part B; NPI, Neuropsychiatric Inventory; 8-OHDG, 8-hydroxy-2′-deoxyguanosine; ox-LDL, Oxidized Low-Density Lipoprotein; MDA, Malondialdehyde; SOD, Superoxide Dismutase; IL, Interleukin; CRP, C-Reactive Protein; FIB, Fibrinogen; CGA-NPI, Comprehensive Geriatric Assessment—Neuropsychiatric Inventory; NPI-NH, Neuropsychiatric Inventory—Nursing Home version; DEI, Dementia Elderly Integration Scale; MMSE-J, Japanese Mini-Mental State Examination; RBANS-J, Repeatable Battery for the Assessment of Neuropsychological Status—Japanese version; SCD, subjective cognitive decline; MCI, mild cognitive impairment; S-GDS, Short Geriatric Depression Scale; CDR, Clinical Dementia Rating; MMT, Manual Muscle Testing; K-MBI, Korean Modified Barthel Index; VFSS, Videofluoroscopic Swallowing Study.

## Data Availability

No new data were created or analyzed in this study. Data sharing is not applicable to this article.

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
