# Peer review of "Current Utilization and Research Status of the Herbal Medicine Guibi-Tang and Its Variants for Cognitive Impairment: A Scoping Review"

_nutrients, 2025, doi:10.3390/nu17213365_

Round 1

Reviewer 1 Report

Comments and Suggestions for Authors

Dear authors,

Congratulations on your work.  

This manuscript provides a thorough review of the latest research on Guibi-tang (GBT) and its variants (KGBT) for cognitive impairment, including conditions like MCI, Alzheimer’s disease, and post-stroke cognitive decline. The authors compile a broad range of clinical and preclinical data, offering valuable insights into potential mechanisms and future research directions. The topic is both relevant and timely, given the increasing interest in integrative approaches to neurocognitive disorders. The manuscript is well-organized and clearly written.

While the discussion provides a comprehensive overview of the findings by linking clinical outcomes to mechanistic insights from preclinical studies, it falls short in thoroughly exploring the implications of heterogeneity. Although heterogeneity is acknowledged as a limitation, it could be examined more closely to better understand how this variation might influence outcome differences, potentially decrease observed effect sizes, or impede the development of standardized treatment protocols.

The conclusions emphasize the preliminary nature of the current evidence and stress the need for larger, multicenter, well-controlled trials that include objective biomarkers.

Based on the points above, I suggest accepting the manuscript with minor revisions.

Reviewer 2 Report

Comments and Suggestions for Authors

The manuscript entitled " Current Utilization and Research Status of the Herbal Medicine Guibi-tang and Its Variants for Cognitive Impairment: A Scoping Review" by Kim and co-workers, is an interesting contribution to the field of multi-herb formulas (in particular GBT and KGBT) prescribed for memory loss, insomnia, and fatigue in East Asian countries. Preclinical studies based on these compounds suggest that they may be of interest as neuroprotective agents, but there is no definitive clinical evidence in this regard.

In this review, the authors have analyzed the available research on GBT and KGBT on the cognitive spectrum. Using the Arksey-O'Mal framework law and the PRISMA-ScR guidelines, we searched seven databases (MEDLINE, 20 Embase, Cochrane Library, China National Knowledge Infrastructure, ScienceON, Sco-21 pus, Citation Information by National Institute of Informatics) for human studies evaluating GBT or KGBT at different stages of cognitive deterioration, from subjective to secondary to stroke. For this reviewer, the use of the China National Knowledge Infrastructure database is of great interest since this type of compounds is more widespread in Eastern cultures.

The authors noted that in studies that meet the GBT or KGBT methodological criteria (developed in South Korea, Japan, and China), they produced modest but consistent improvements in global cognition (Mini-Mental State Examination/Montreal Cognitive Assessment), memory domains, activities of daily living, and neuropsychiatric symptoms of some of the cohorts.  without any major adverse events. The enormous differences in the administration times of the compounds are striking. However, as the authors point out in the conclusions, standardized, multicenter, controlled trials are needed to clarify and scientifically prescribe GBT or KGBT.

Overall, the Review is well written, informative and of interest to all bio-sanitary professionals interested in multi-herb formulas; surely the manuscript is more interesting for East Asian countries where there is an ancient culture in the use of these formulas.

Reviewer 3 Report

Comments and Suggestions for Authors

Kim et al. conducted a scoping review on Guibi Tang and Kami Guibi Tang for cognitive impairment. It is very intriguing and well done.

There are some typos.

Line 50, the reference number should be placed at the end of the sentence.

Line 108, the reference number should be placed befor dot.

Line 653 and Line 654, the reference 24 and reference 25 should be the same.

Reviewer 4 Report

Comments and Suggestions for Authors

Current Utilization and Research Status of the Herbal Medicine Guibi-tang and Its Variants for Cognitive Impairment: A Scoping Review

The manuscript provides a scoping review on the use of Guibi-tang (GBT) and its variants for cognitive impairment. While the topic is timely and potentially relevant, several methodological and conceptual weaknesses need to be addressed.

The manuscript's writing style needs to be improved. For example, some sections are vague. This is already evident in the abstract, which mentions multimodality without giving examples.

There are also formatting errors in the text. For example, on line 50, you can see a reference in the middle of a word.

Line 61-72. The study matrix raises questions. To begin with, it refers to a mixture of several herbs. Are there established proportions, or does each consumer/seller mix them to their liking? Furthermore, the variability in composition within a single species varies greatly depending on soil and climate factors or treatment, among others. Is it guaranteed that all formulations have the same composition? Clinical application is discussed at the end of this paragraph. Has a drug been developed from these raw materials? The source, quality, and standardization of the herbal raw materials are not described. Without details on origin, preparation methods, or chemical consistency, it is impossible to assess reproducibility and clinical reliability. This is particularly important in herbal research, where differences in raw material sourcing can significantly alter pharmacological effects.

Line 85. Small trials. What is small?

Section 2. There are some unnecessary elements in this section and incorrect terms. For example, there is no need to define what this type of review consists of, and the name of the review should be corrected.

The research question is framed too broadly and lacks precision. “What studies have applied GBT or its modified forms to cognitive impairment?” is not sufficiently specific for a scoping review, especially when highly heterogeneous variants are included.

Authors often use first-person verbs. This is inappropriate for scientific reviews / articles.

The search strategy is not transparent. While the databases are listed, the exact search strings and inclusion/exclusion criteria are vague. The appendix is mentioned, but the main text should summarize the terms and logic applied.

Section 2 contains a lot of information that isn't relevant to the article, while essential information for this type of article is missing. For example, specific exclusion criteria, number of publications reviewed, years reviewed, etc.

The section states that “systematic reviews were included when relevant,” which is inappropriate for a scoping review. Mixing primary studies and reviews without clear separation introduces bias.

The description of study selection is repetitive and lacks clarity. For example, it is unclear how disagreements were resolved beyond “discussion” and how many records were excluded at each stage.

The paper repeatedly refers to “GBT” and “KGBT,” but most of the included studies used substantially modified formulas, with additions and subtractions of core ingredients (see Table 2). This raises doubts about whether the authors are evaluating a consistent intervention or a heterogeneous set of herbal mixtures. The pharmacological rationale for pooling these formulations is not adequately justified. Moreover, results from randomized controlled trials, observational studies, and even single case reports are presented together in the same flow, without sufficient distinction. This makes it difficult for readers to assess the strength of the evidence.

The results section tends to highlight improvements in cognitive outcomes while minimizing or glossing over studies that reported neutral or negative findings. For instance, Kim et al. [13] clearly found no cognitive benefit in moderate Alzheimer’s disease, yet this result is presented almost as an exception rather than a central limitation of the evidence. A balanced review should give equal weight to studies that failed to show efficacy (e.g., Kim et al. [13] reporting no cognitive advantage in AD is downplayed).

The section often emphasizes global cognitive improvements measured by MMSE or MoCA but gives little attention to cases where secondary or domain-specific tests failed to show significant change. For example, some studies reported improvements in certain subdomains but not in overall cognition, yet this nuance is lost in the current summary. Similarly, biochemical or imaging findings are reported without adequate explanation of their clinical significance, which can mislead readers into overestimating their importance.

In several instances, numerical values are reported selectively or incompletely.

While the authors note that all studies were conducted in East Asia, the implications of this are not properly integrated into the results. Cultural, dietary, and genetic factors may significantly influence both tolerability and efficacy of herbal preparations.

The tables are nowhere near the quality required for a review article. For starters, they are very brief. Then there is some poorly formatted information. For example, in the first column of Table 1, the references are mixed with the reference itself in a different format. If the authors want to keep the year (which would not be relevant if the Materials and Methods section mention the years chosen for bibliography), it should be in a separate column from the reference itself. Furthermore, the reference should be in the last column. Furthermore, this table provides little information. The only relevant information is the sample size and the dose, but the results of each of these studies, which is what is truly interesting, are not discussed. Something similar happens in Table 2.

Including a discussion section when the authors do not pause to analyze anything is quite inappropriate. The authors simply analyze the bibliography without examining the limitations or providing a new critical perspective.

There is a section 6 of patents that has absolutely nothing.

Not a single figure other than the one describing how the articles were analyzed? It would be interesting to have at least one that discusses the main mechanisms responsible for the preparation's bioactivity or the main molecules responsible for its bioactivity. This is further evidence that the manuscript still has a huge amount of work to do.

Comments on the Quality of English Language

The English could be improved to more clearly express the research.

Round 2

Reviewer 4 Report

Comments and Suggestions for Authors

The authors were previously asked to make and consider a substantial number of changes and suggestions, but very few of these have actually been addressed in the revised version. In addition, nearly half of the current manuscript consists of a series of tables that are difficult to read due to the lack of proper formatting. It is challenging to compare the data across columns, which significantly affects readability and interpretation. Overall, the manuscript still shows a considerable lack of work and scientific content, and it does not yet meet the standards required for publication.

Comments on the Quality of English Language

 The English could be improved to more clearly express the research.
